# NFCMTL: Auto NailFold Capillaroscopy through a Multi-Task Learning Model

Yingke Ding[1,3][0009−0006−3618−9492], Jiankai Tang[1][0009−0009−5388−4552],
Wanying Mo[1][0009−0009−9426−2426], Tianruo Rose Xu[4][0009−0005−7462−5788],
Yuanchun Shi[1,2], and Yuntao Wang[1]⋆

[1] Tsinghua University, Beijing, China
[2] Qinghai University, Qinghai, China
[3] University of Washington, Seattle, Washington, United States
[4] Cornell University, Ithaca, New York, United States
{dyk21,tjk24,mwy21}@mails.tsinghua.edu.cn, tx88@cornell.edu
{shiyc,yuntaowang}@tsinghua.edu.cn

**Abstract.** Nailfold capillaroscopy is a non-invasive technique for assessing microvascular health by visualizing capillaries in the nailfold, playing a key role in diagnosing vascular and autoimmune diseases. We propose a novel machine learning approach for nailfold analysis, introducing an advanced multi-task learning model that jointly performs capillary segmentation, classification, and keypoint detection within a unified architecture. Using a large public dataset with reorganized keypoint annotations, our approach improves precision and efficiency in feature detection while simplifying the conventional multi-stage pipeline. By leveraging multi-task optimization, the model achieves state-of-the-art performance comparable to existing methods. This work advances nailfold imaging by providing an accurate, streamlined solution for automated, non-invasive microvascular diagnostics. Code is available at https://github.com/thuhci/NFCMTL.

**Keywords:** Nailfold Capillaroscopy · Multitask Learning · Vision Transformer.

## 1 Introduction

NailFold Capillaroscopy (NFC) is a non-invasive imaging modality used clinically to assess the health of the microcirculatory system by visualizing capillary structures near the surface of the skin, particularly at the human finger nailfold area [8,4,2]. This imaging technique provides critical insights into capillary morphology, making it indispensable in diagnosing and monitoring a range of autoimmune and vascular conditions, including Systemic Sclerosis (SSc) [13,23] and Raynaud's phenomenon [22,18]. Moreover, emerging research indicates that NFC abnormalities might correlate closely with metabolic disorders such as diabetes [17,24], thus further extending its diagnostic relevance.

---

⋆ The corresponding authors.

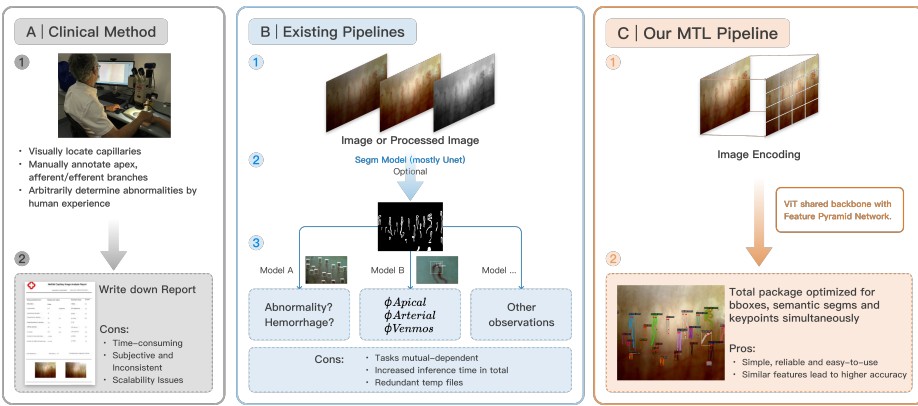

**Fig. 1.** Traditional method for NFC (**A**) requires significant clinician intervention. While existing deep learning approaches (**B**) allow clinicians to process images through multiple separate models, our proposed multi-task learning model (**C**) integrates key tasks into a unified model that produces comprehensive capillary image analysis in a single operation. Note: the A①image is obtained from [10].

In traditional clinics, NFC examinations involve capturing microscopic images at approximately ×200 magnification, followed by detailed manual analysis. Specialists visually inspect these images for morphological abnormalities, delineate capillary boundaries, classify capillary morphology, and measure parameters such as apical, arterial, and venous diameters. They then compare these morphological and quantitative findings to reference criteria or rely on clinical experience to identify abnormalities indicative of disease (Figure 1 (**A**)). While effective, this manual assessment is inherently *subjective, labor-intensive, time-consuming*, and highly *dependent* on clinician expertise, potentially leading to diagnostic variability, inconsistent image interpretation, and delayed or inaccurate clinical decisions [1]. Additionally, the manual approach significantly strains clinical resources, restricting NFC's broader accessibility and clinical adoption.

The application of machine learning in nailfold capillaroscopy is advancing accurate automated diagnosis. A common starting point in this process is segmentation, which outlines targeted capillaries in input images. While not strictly necessary for morphological estimation or parameter calculation, segmentation enhances deep learning pipelines by improving capillary localization [3]. Neural networks like U-Net [21], Mask-RCNN [9] and their variants [16,19] are widely used in NFC segmentation.

Beyond segmentation, capillary classification also plays a crucial role in identifying different capillary types, such as normal, abnormal, or those with conjunctions or anastomoses [7,29]. CNN-based approaches have proven effective for this task. Meanwhile, the quantification of capillary parameters, including density, loop width, and arterial/venous length, is essential for diagnosing dis-

eases like systemic sclerosis, lupus, and rheumatoid arthritis. Some studies favor traditional computer vision or mathematical methods for this analysis [12,7].

Recent viral NFC studies also favor keypoint-based quantitative analysis. For example, Tello et al. [7] employed stacked DenseNet for two-stage capillary parameter estimation, achieving 88% accuracy at a confidence threshold of 0.50. Zhao et al. [29] combined Mask-RCNN with a matching algorithm, reporting an apical diameter MAE of 1.674 pixels and RMSE of 2.023 pixels. Integrating keypoint estimation into NFC analysis enhances accuracy and efficiency.

Despite successes in individual NFC tasks, existing methods fail to **simultaneously** predict multiple tasks. Similar medical imaging studies, such as retinal fundus [28] and skin lesions [25], have demonstrated strong connections between related tasks like classification and segmentation. General imaging applications, including human pose estimation, also indicate a strong link between keypoint estimation and segmentation [6].

To bridge the gap in existing NFC research, we introduce a novel Multi-Task Learning (NFCMTL) strategy, depicted in Figure 1**(C)**, that integrates capillary *semantic segmentation*, *keypoint detection*, and *classification* into a single unified model. Leveraging a Multiscale Vision Transformer (MViT) backbone with a Feature Pyramid Network (FPN), our model uses a specialized loss function to optimize task predictions simultaneously. Evaluations on the ANFC dataset [29] demonstrate balanced performance improvements across tasks. Ablation studies further confirm precision gains from task unification, achieving sub-pixel accuracy ($< 1$ pixel error) in downstream capillary parameter estimations.

Our contributions are summarized as follows: **1)** We propose the first reliable multi-task learning model for NFC image tasks, simultaneously performing precise segmentation, classification, and keypoint estimation. **2)** We introduce the **MViT-FPN** model, which outperforms existing approaches in NFC imaging tasks. **3)** Through extensive experiments, NFCMTL demonstrates superior performance in capillary entity estimation and parameter computation, especially when keypoint detection is jointly learned.

## 2    Method

### 2.1    Dataset, annotations, and keypoints definition

The dataset comprises $N$ clinician-selected RGB capillaroscopy images of size $W \times H \times 3$, collected from multiple participants. Each image $\mathcal{M}_j$, where $j \in 1, 2, ..., N$, contains several visible capillaries and optional hemorrhages. The capillaries in $\mathcal{M}_j$ are represented as a list of entities $\{\mathcal{E}_1^{(j)}, \mathcal{E}_2^{(j)}, \ldots, \mathcal{E}_w^{(j)}\}$, where $w$ is the number of annotated capillaries in $\mathcal{M}_j$.

A capillary entity $\mathcal{E}_i^{(j)}$ can now be written as a tuple of four components:

$$\mathcal{E}_i^{(j)} = \left( \mathcal{S}_i^{(j)}, \mathcal{B}_i^{(j)}, \mathcal{Q}_i^{(j)}, \mathcal{P}_i^{(j)} \right) \tag{1}$$

In Equation 1, $\mathcal{S}_i^{(j)}$ denotes the segmentation polygon, $\mathcal{B}_i^{(j)}$ refers to the bounding box, $\mathcal{Q}_i^{(j)}$ corresponds to the classification label and $\mathcal{P}_i^{(j)}$ represents the

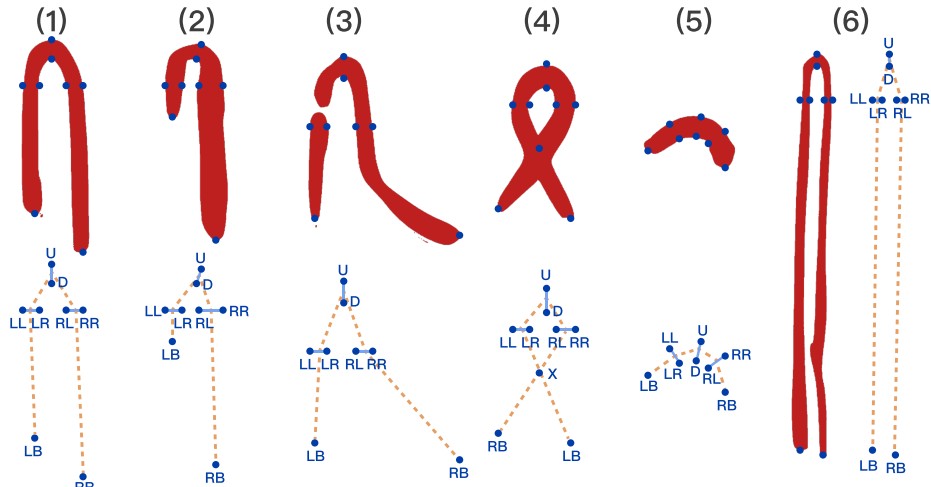

**Fig. 2.** Different types of nailfold capillaries examined in this research. **(1)** Normal capillary displaying the classic inverted-U shape. **(2)** Abnormal capillary where the afferent limb (left portion) is shorter than the efferent limb (right portion). **(3)** Abnormal capillary characterized by a non-linear efferent limb. **(4)** Abnormal capillary exhibiting a conjunction or anastomosis. **(5)** Abnormal capillary with both afferent and efferent limbs shorter than typical length. **(6)** Abnormal capillary with both afferent and efferent limbs longer than typical length.

set of keypoints. In this study, we define a fixed set of 9 keypoints, which include the up ($U$), down ($D$), left-left ($LL$), left-right ($LR$), right-left ($RL$), right-right ($RR$), left-bottom ($LB$), right-bottom ($RB$) and the optional conjunction ($X$). The $U$ and $D$ points are chosen from the capillary apex, $LL$ and $LR$ are selected from the arterial limb, while $RL$ and $RR$ are selected from the venous limb. All keypoints in a capillary entity are represented as one-hot binary masks of size $m \times m$, where each mask encodes a specific anatomical landmark and a spatial softmax over the $m^2$ grid is applied to predict the most probable keypoint location during inference, as described in [9]. Note that all capillaries in this study must contain the 8 essential keypoints, with the exception of the optional conjunction point. Hemorrhages class do not contain any keypoints. To facilitate better representation of this information, we adopt the MS COCO format [15] for annotations, assigning a visibility flag to each keypoint. Figure 2 illustrates various keypoint annotations in this research.

### 2.2   Model Architecture

**Overview** The architecture of the proposed model is illustrated in Figure 3. While the final tasks share close relationships, they necessitate distinct prediction strategies due to the intricate characteristics of nailfold capillaries — small size, subtle visual features, occlusions, and irregular shapes. Consequently, the

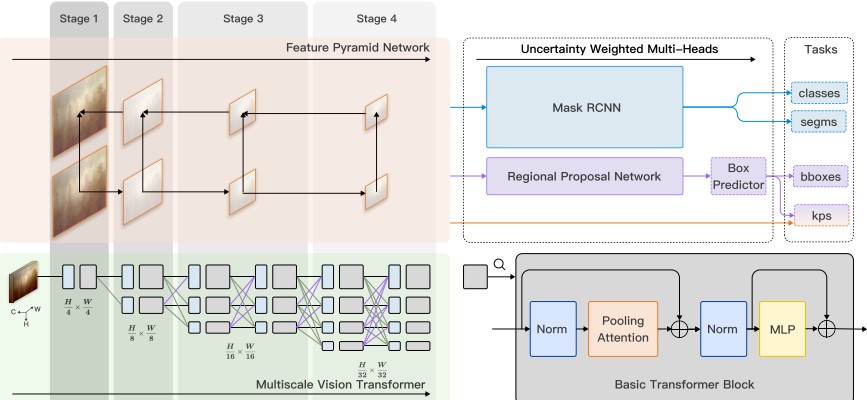

**Fig. 3.** Overview of the proposed architecture. The model integrates a Multiscale Vision Transformer backbone with a Feature Pyramid Network to capture both fine-grained details and high-level structural context of the capillaries. The multi-scale features are fed into task-specific heads built on Mask R-CNN and Region Proposal Network to perform segmentation masks, classification, and keypoint heatmap tasks.

model must effectively capture both low-level visual details and high-level structural relationships. We adopt the Multiscale Vision Transformer (MViT)[5,14] to address these challenges — we encode each input into multi-scale square patches through adaptive upsampling and downsampling operations. These multi-scale patches align naturally with the four-stage Feature Pyramid Network (FPN), subsequently feeding into dedicated task-specific heads — Mask-RCNN [9] and Region Proposal Network (RPN) [20], where a custom total loss function is applied to optimize multi tasks, resulting in a comprehensive prediction of capillary representations.

**Backbone** Firstly, all input nailfold capillary images are adopted paddings for a square shape. We then segment the image into $P \times P$ patches. For ViT-based encoding, the model extract features to encode each patch into a multi-scale feature map $\mathbf{F} \in \mathbf{R}^{\frac{H}{P} \times \frac{W}{P} \times D}$. To create multiple feature maps at different scales as used in FPN, we perform recursive down-sampling and up-sampling using convolutional layers with stride $s \sim \{4, 8, 16, 32\}$. Later in the FPN module, we return the feature map to its original resolution by upsampling recursively.

**Attention Module** Just like other ViT systems, the attention scores between each pair of patches are computed by taking the dot product between the query $Q$, key $K$ and value vectors $V$. Let $X = [x_1, x_2, \ldots, x_N]$ be the sequence of input patch representations, the three vectors can be calculated with learnable weight matrices $Q = XW_Q, K = XW_K, V = XW_V$.

The resulting scores are scaled by $\frac{1}{\sqrt{d}}$, $d = \frac{H}{P} \times \frac{W}{P} \times D$ to stabilize gradients during training. These scores are passed through softmax for weights.

**ROI Heads** The downstream ROI heads refine FPN-generated proposals and make final predictions for the capillary tasks with the structure of Mask-RCNN and RPN. The model utilizes ROI Align adopted from the Mask R-CNN framework to extract features from each proposal. Following this, the model performs bounding box regression to adjust coordinates and classifies capillaries using a softmax function. To predict the keypoint heatmap, we use a top-down method that takes the predicted bounding boxes with FPN output features to generate the heatmap. Capillary semantic segmentation is achieved by piping FPN's output features into Mask R-CNN, which then uses five $256 \times 256$ Conv2d heads to predict five distinct pixel-level classes.

### 2.3   Task-specific Loss and Multitask Loss

**Classification Loss** Let $\hat{y}$ be the predicted probability distribution (from the final softmax layer), and $y$ be the true class label (a one-hot encoded vector). The classification loss is hereby $\mathcal{L}_{\text{class}} = -\sum_{c=1}^{C} y_c \log(\hat{y}_c)$, where $C$ is the number of classes, $y_c$ is the true label for class $c$, and $\hat{y}_c$ is the predicted probability for class $c$.

**Segmentation Loss** We use the *Dice Loss* for segmentation $\mathcal{L}_{\text{segm}} = 1 - \frac{2\sum_i \mathcal{S}_i g_i}{\sum_i \mathcal{S}_i + \sum_i g_i}$. Here $\mathcal{S}_i$ is the predicted segmentation mask at pixel $i$, and $g_i$ is the ground truth segmentation mask at pixel $i$.

**Bounding Box Loss** For the bounding box regression task, we use the *Smooth L1 Loss*. Let $\mathcal{B}_i^{(j)} = [x_{\min}, y_{\min}, x_{\max}, y_{\max}]$ be the predicted bounding box for the $i$-th capillary, and $\mathcal{G}_i^{(j)} = [g_{\min}, h_{\min}, g_{\max}, h_{\max}]$ be the ground truth bounding box. The bounding box loss is defined as $\mathcal{L}_{\text{bbox}} = \frac{1}{4}\sum_i \left(\left|\mathcal{B}_i^{(j)} - \mathcal{G}_i^{(j)}\right|^2\right)$.

**Keypoint Loss** For the keypoint detection task, we use the *cross-entropy loss*: $\mathcal{L}_{\text{kp}} = \frac{1}{N}\sum_{n=1}^{N}\sum_{k=1}^{K}\sum_{i=1}^{S}\sum_{j=1}^{S} -g_{n,k,i,j}\log(\hat{g}_{n,k,i,j})$. The predicted keypoints are represented as a tensor of logits of shape $(N, K, S, S)$, where $N$ is the batch size, $K$ is the number of keypoints, and $S$ is the size of the keypoint heatmap. The ground truth keypoints are converted into heatmaps.

**Total Loss** Since all tasks here are jointly optimized, weighting strategies are vital for the final optimization. Brought the idea from [11], we use uncertainty weighting to balance these tasks: $\mathcal{L}_{\text{total}} = \mathcal{UW} \otimes \{\mathcal{L}_{\text{class}}, \mathcal{L}_{\text{segm}}, \mathcal{L}_{\text{bbox}}, \mathcal{L}_{\text{kp}}\}$.

**Table 1.** Summary of evaluation results

| Task | Subtask | Metrics | Ours | ANFC [29] | CAPI [7] | Mask-RCNN [9] |
|------|---------|---------|------|-----------|----------|---------------|
| Segm. | Capillary Mask | Sens.↑ | **0.827** | 0.653 | — | 0.820 |
| KP. | Venous Diameter | MAE↑ | 1.813 | **0.989** | 1.274 | 1.794 |
| KP. | Arterial Diameter | MAE↓ | **0.825** | 0.849 | 0.856 | 1.351 |
| KP. | Apical Diameter | MAE↓ | **0.321** | 1.674 | 0.575 | 1.047 |
| Class. | Abnormal State | Accuracy↑ | **0.885** | 0.800 | 0.747 | 0.839 |

The test set includes 61 images from various distinct subjects, ensuring no overlap between the subjects in the training and test datasets. Classification metric is calculated per image level and keypoint task's metric is calculated at the pixel level. For the ANFC [29] we report the original results from the paper. For the CAPI [7] and Mask-RCNN [9] results, we implemented the method as described in the original paper and evaluated it on the dataset used in this work.

## 3   Experiment and Discussion

**Dataset**  We utilized a public dataset of nailfold videocapillaroscopy images from [29]. This dataset contains 321 high-quality capillaroscopy recordings from 68 participants with a microscope of around ×200 magnification. By submitting Biometrics Dataset Release Agreement, we obtain both the raw image data and its Labelme [26] formatted annotation and converted them to COCO format as described in the above section.

**Implementation**  Model training was conducted on an NVIDIA GeForce RTX 4090 GPU (24GB) under Ubuntu 24.04, using Python 3.11.11, PyTorch 2.6.0, and Torchvision 0.21.0. Input images were uniformly resized to $1024 \times 1024$ and augmented via flipping, cropping, resizing. In addition to those augmentations, we applied photometric transformations like random brightness, contrast, and hue shifts to mimic the highly variable lighting conditions found in outpatient clinics. We select MViTv2-T as the initial checkpoint for its strong performance and lightweight architecture. Optimization employed AdamW with an initial learning rate of 1.6e-4, incorporating linear warm-up and scheduled decay at 52,500, 62,500, and 67,500 iterations. Due to GPU memory constraints, the batch size was fixed at 4. An 80:20 train-test split was applied at the subject level to prevent image overlap across sets. Although it has been suggested that Detectron2 [27] may not be ideal for certain experiments [7], we successfully engineered the core codebase using this platform.

**Evaluation Metrics**  In this research, different metrics are picked for each task, as shown in Table 1. For each task, we conduct experiments using a five-fold setup and report the aggregated results across the folds.

**Results**  For segmentation, we evaluate pixel-level sensitivity and obtain an average result of 0.827, outperforming ANFC [29] by demonstrating improved

**Table 2.** Different task combination and evaluation results

| Tasks | Segm (Sens.) | Class (Acc.) | MAE$_{apex}$ | MAE$_{venous}$ | MAE$_{arterial}$ |
|---|---|---|---|---|---|
| Segm+Class | 0.79 | **0.905** | — | — | — |
| Segm+KP | 0.82 | — | 1.823 | 2.235 | 0.942 |
| Class+KP | — | 0.819 | 0.632 | 2.033 | 1.725 |
| **All tasks** | **0.83** | 0.885 | **0.321** | **1.813** | **0.825** |

segmentation performance. Regarding CAPI [7], since the original method does not address the segmentation task, we do not include it in this comparison.

For the classification task, we assess results at the *whole-image level*. Our assumption is that if an NFC image contains a single abnormal capillary, the entire image is classified as abnormal. Similarly during evaluation, the inferred classifications of individual capillaries are aggregated to determine the overall image classification. Based on this assumption, we hit an 88.5% accuracy, 89.7% precision, 86.8% recall and 88.2% F1 score.

For the keypoint detection task, we assess model performance through a downstream task: capillary parameter estimation. Specifically, the venous diameter is computed as the Euclidean distance between keypoints $LL$ and $LR$, the arterial diameter from $RL$ and $RR$, and the apical diameter from $U$ and $D$. The mean absolute error (MAE) is calculated as the average difference between the predicted and ground truth diameters for all capillaries at the image level. As shown in Table 1, the model demonstrates strong performance, particularly in estimating arterial and apical diameters.

**Ablation Study** Table 2 shows the impact of different task combinations on segmentation, classification, and keypoint performance. The final combined task performs well over major tasks, proving unified optimization works as expected.

## 4    Conclusion

In this paper we introduce a unified multi-task model for nailfold capillaroscopy that merges segmentation, keypoint detection, and capillary classification. Using a Multiscale vision transformer backbone with Mask R-CNN heads, the model delivers more precise capillary geometry and measurements for capillary parameters. This streamlined approach enhances automated NFC analysis, promising better clinical diagnostics and supporting future multi-task NFC research.

**Acknowledgments.** This work is supported by the National Key R&D Program of China under Grant No. 2024YFB4505500 & 2024YFB4505503, the National Natural Science Foundation of China under Grant No. 62472244, the Qinghai University Research Ability Enhancement Project (2025KTSA05), the foundation of National Key Laboratory of Human Factors Engineering under Grant No. HFNKL2024W06, the Tsinghua University Initiative Scientific Research Program, and Undergraduate Education Innovation Grants, Tsinghua University.

**Disclosure of Interests.** The authors have no competing interests to declare that are relevant to the content of this article.

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
