# OpenReview forum: "NFCMTL: Auto NailFold Capillaroscopy through a Multi-Task Learning Model"
_MICCAI.org/2025/Workshop/MSB_EMERGE — MSB EMERGE 2025 Oral_

### Official Review · Reviewer_mzKw · 2025-07-09

**Recommendation:** 4
**Confidence:** 3

**Clarity:**

The paper is clear and well-written, with minor areas for improvement in clarity

**Feedback:**

1. Please provide comparative experimental results to demonstrate how the proposed method improves in terms of parameter efficiency.
2. Conducting external validation using datasets other than the one used for training is recommended to better assess the generalizability of the model.
3. Methods mentioned in Section 2.2, such as ROI Align, should be clearly stated as components adopted from Mask R-CNN, to clarify their source.

**Justification:**

The proposed approach has its value in that it streamlines the traditionally disjointed NFC pipeline, offering a more accurate and clinically viable solution for microvascular assessment. A few modifications and experimental backups would enhance the value of this work.

**Reproducibility:**

Sufficient amount of details available for reproducing the main results, and open access is provided (or promised upon acceptance) to source code and/or data

**Strengths:**

1. The objective of the paper is clearly stated—it presents the first unified model that simultaneously performs multiple tasks for nailfold capillaroscopy analysis.
2. The implementation details are well-documented in Section 2, with a clear description of the model architecture and loss functions.
3. The experimental results show promising performance across all tasks, indicating the potential clinical relevance of the proposed approach.

**Summary:**

This paper presents NFCMTL, a novel multi-task learning framework for automated nailfold capillaroscopy analysis, integrating capillary segmentation, keypoint detection, and classification into a unified vision transformer-based architecture. The authors employ a Multiscale Vision Transformer backbone with Feature Pyramid Networks and task-specific heads to enhance performance across tasks. Quantitative evaluations demonstrate significant improvements over preceding research.

**Weaknesses:**

1. While the paper claims superior performance in capillary parameter computation, no quantitative results are provided to support this. Comparative results should substantiate this claim.
2. The dataset used (321 images from 68 patients) appears relatively small, raising concerns about potential overfitting. It would be helpful to discuss any measures taken to mitigate this risk.
3. The implementation method itself is not novel, as the model simply adopts a base transformer block without further architectural innovation.

---

### Official Review · Reviewer_eYCT · 2025-07-09

**Recommendation:** 4
**Confidence:** 4

**Clarity:**

The paper is generally clear but has some clarity issues that could be addressed with moderate revision

**Feedback:**

-	Evaluation: What did you change when repeating the experiment 5 times? Reporting the standard deviation or mentioning it is below a certain value would have been relevant
-	The second line of Table 1, MAE goes with an arrow down
-	Be careful of long acronyms that are hard to read; they can complicate your message (e.g. title of Fig. 3)
-	It is not clear if the SOTA reporting measures are from an individual model for each task or a similar multi-task learning approach.

**Justification:**

The proposed method addresses a relevant problem and shows improved performance over state-of-the-art approaches, which justifies a weak accept. However, the lack of novelty in the implementation limits the overall contribution.

**Reproducibility:**

Sufficient amount of details available for reproducing the main results, and open access is provided (or promised upon acceptance) to source code and/or data

**Strengths:**

- The paper tackles a relevant problem for the community
- Paper is reproducible: An Extensive description of the implementation details is provided, and the code is available
- A relevant ablation study is conducted

**Summary:**

In this paper, the authors proposed an multi-task learning model for NailFold Capillaroscopy (NFC) to solve three different tasks (capillary semantic segmentation, keypoint detection, and classification) using only one model.  They demonstrate the benefit of their approach on a public dataset across all tasks against  specific model for each task.

**Weaknesses:**

- Uncertainty weighting loss: No explanation why it is relevant to use the uncertainty weighting strategies to combine all of the losses. An ablation study of the impact of this would have been relevant.
-	Metrics reported: only one metric per task is reported. For the classification task, accuracy can be strongly impacted by class imbalance.
- Limited novelty in the proposed approach; key components appear to be based on existing methods with minimal innovation

---

### Official Review · Reviewer_psZc · 2025-07-11

**Recommendation:** 4
**Confidence:** 4

**Clarity:**

The paper is generally clear but has some clarity issues that could be addressed with moderate revision

**Feedback:**

See weaknesses + :
- Show some visual examples of results.
- Mention any limitations or next steps for clinical use.

**Justification:**

The paper addresses a relevant clinical task with a practical multi-task approach, but limited baseline comparisons, architectural complexity, and lack of clarity of some design choices, e.g., hyper-parameter tuning, reduce its impact

**Reproducibility:**

Sufficient amount of details available for reproducing the main results, and open access is provided (or promised upon acceptance) to source code and/or data

**Strengths:**

- The idea of combining tasks into one model is practical and likely improves efficiency.
- The work is relevant for diagnosing vascular and autoimmune diseases.
- Using a public dataset with improved annotations makes the results more robust.
- The three tasks—segmentation, classification, and keypoint detection—appear to complement each other well within the joint architecture, leading to improved performance compared to handling them separately.

**Summary:**

This paper introduces a new machine learning model for analyzing nailfold capillaroscopy images. The authors propose a multi-task learning approach that combines three tasks, i.e., capillary segmentation, classification, and keypoint detection into a single model, and show that their method performs on par or better than existing approaches. Code is made publicly available.

**Weaknesses:**

- Only one baseline is used for comparisons, which limits the strength of the performance claims
- Some large VLMs (e.g., InternVL) can perform grounding, segmentation, and classification. How do they perform in comparison? Is it worth pursuing fine-tuning of these models, rather than training other architectures from scratch?
- Generalization to other datasets or imaging conditions is not discussed
- Some figures (e.g., Fig. 3) and tables (e.g., Table 1, 2) would benefit from being placed consistently at the top or bottom of the page for better readability
- Figure captions should be more informative and self-contained. For example, a caption like "NFCMTL's MViT-FPN Architecture" doesn’t clearly explain what the figure shows
- There appears to be a mistake in Table 1: the arrow for MAE in the second row should point downward, as lower MAE indicates better performance
- Is the use of Mask R-CNN–style ROI heads necessary for this domain, or could a more unified architecture achieve similar results with less complexity?
- The use of uncertainty weighting to balance the multi-task losses is neither clearly motivated nor supported by ablation or experimental results